# Pharmacokinetics of Levofloxacin Entrapped in Non-Ionic Surfactant Vesicles (Niosomes) in Sprague Dawley Rats

**DOI:** 10.3390/pharmaceutics17020275

**Published:** 2025-02-18

**Authors:** Amusa S. Adebayo, Satish Jankie, Jenelle Johnson, Lexley Pinto Pereira, Kafilat Agbaje, Simeon K. Adesina

**Affiliations:** 1Department of Pharmaceutical Sciences, College of Pharmacy, Howard University, 2300 4th Street NW, Rm 309, Washington, DC 20059, USA; kafilat.agbaje@bison.howard.edu (K.A.); simeon.adesina@howard.edu (S.K.A.); 2School of Pharmacy, Faculty of Medical Sciences, University of the West Indies, St. Augustine, Trinidad and Tobago; satish.jankie@sta.uwi.edu; 3School of Veterinary Medicine, Faculty of Medical Sciences, University of the West Indies, St. Augustine, Trinidad and Tobago; jenelle.johnson2@sta.uwi.edu; 4Department of Paraclinical Sciences, Faculty of Medical Sciences, University of the West Indies, St. Augustine, Trinidad and Tobago; lexley.pintopereira@sta.uwi.edu

**Keywords:** niosomes, levofloxacin, pharmacokinetics, Sprawgue Dawley rats

## Abstract

**Background/Objectives**: Bacteria are becoming increasingly resistant to levofloxacin and other fluoroquinolones. Previously, drug loading in colloidal carriers has shown enhanced penetration into and retention in bacterial cells. However, the mechanism of levofloxacin niosomes’ bio-disposition in rats has not been reported. This study investigated the pharmacokinetics (PK) of optimized levofloxacin niosomes following intraperitoneal injection into Sprague Dawley rats. **Methods**: Formulation and processing variables settings were determined using DoE Fusion One software. The resulting data input into the Optimizer module provided niosome formulation for in vivo study in Sprague Dawley rats. Each group of rats (n = 6) was injected intraperitoneally with either conventional levofloxacin or its niosomes at equivalent doses of 7.5 mg/kg/dose. Blood samples were collected via tail snip and analyzed using a validated HPLC method. The plasma–time data were fed into the Gastroplus software (Simulations Plus, CA) and used to model levofloxacin PK. **Results**: Niosomes for in vivo study had a mean hydrodynamic diameter of 329.16 nm (±18.0), encapsulation efficiency (EE) of 30.74%, Zeta potential of 21.72 (±0.54), and polydispersity index (PDI) of 0.286 (±0.014). Both the Akaike and Schwarz criteria showed levofloxacin niosomes and conventional drug formulation obeying one- and two-compartment PK models, respectively. Thus, formulation in niosomes altered levofloxacin biodistribution by concentrating the drug in the vascular compartment. **Conclusions**: Niosome encapsulation of levofloxacin altered its biodistribution and pharmacokinetic profile, possibly by protecting i.p. levofloxacin en route into plasma, and significantly enhanced its plasma concentration with enhanced potential for treating intravascular infections.

## 1. Introduction

Niosomes are non-ionic surfactant vesicles usually stabilized with cholesterol and charge-inducing agents. The microscopic lamellar structure formed from the admixture of non-ionic surfactant and cholesterol has the ability to accommodate drugs of various molecular nature with a wide range of solubilities. Niosomes are similar to liposomes in chemical structure and physical properties [1]. Unlike liposomes, niosomes are osmotically active and stable in physiologic fluids and have greater potential for increasing the stability of entrapped drugs. The surfactants used for niosome construction are biodegradable, biocompatible, and non-immunogenic, and the formulation requires no special handling or storage conditions [2,3]. Therefore, niosomes are growing in therapeutic applications for the delivery of a wide range of drugs for various conditions [1,4,5].

Studies have shown that the distribution and localization of colloidal carriers in tissues and cellular and sub-cellular areas parallel the distribution into the common bacteria cells that are responsible for intracellular infections [6]. Topoisomerases, the target enzymes for fluoroquinolones, are in the bacteria cytoplasm, and changes in the permeability of the outer membrane of Gram-negative bacteria cells have been linked with resistance development. It has been hypothesized that encapsulation of drugs in niosomes could provide a stealth effect, enhance internalization by bacteria cells, and improve intracellular delivery to the cytoplasm [7]. Generally, cationic niosomes are internalized faster and to a greater extent than negatively charged neutral particles due to stronger electrostatic attraction to the negatively charged cell membrane [8].

Levofloxacin is the optical S-(-) isomer of the ofloxacin and a Biopharmaceutics Classification System (BCS) class 1 drug, which is highly soluble and highly permeable [9,10]. It has a broad spectrum of activity against Gram-positive, Gram-negative, and atypical organisms, such as Mycoplasma, Legionella, Mycobacterium, and Chlamydia [11]. It is a major component of the antibacterial cocktails of *H. pylori* and is currently being repurposed via molecular modification as an anticancer agent [12]. One of the main mechanisms by which bacteria develop resistance to antimicrobial agents is through the evolution of cellular structures that prevent drug permeation [2,13,14]. Although levofloxacin, a Biopharmaceutics Class 1 drug, is highly soluble and highly permeable [9], bacteria are becoming increasingly resistant to the drug by efflux pump amplification and porin pathway blockade. Its resistance potency loss portends a major setback to the fight against peptic and duodenal ulcers and their potential progression to gastroduodenal carcinoma [15,16,17]. Approaches to levofloxacin formulation for enhanced intracellular delivery involve the use of lipid nanoparticles [18], silica-based mesoporous materials [19], and nanocarriers [20,21].

In a previous study, we investigated formulations of select fluoroquinolones (norfloxacin, ciprofloxacin, gatifloxacin, and levofloxacin) in niosomes [3,22]. In vitro testing against ciprofloxacin-resistant bacterial strains showed at least a two-fold increase in potency [22]. It appears that there is a lack of literature information on the mechanisms of biodistribution of levofloxacin niosomes in vivo. This study was designed to investigate the in vivo pharmacokinetics (PK) of an optimized niosomal formulation of levofloxacin in rats following intraperitoneal administration. The aim was to determine the pharmacokinetic parameters and any impact of niosomes on in vivo distribution of levofloxacin in rats.

## 2. Materials and Methods

Span 60, ultrapure cholesterol, dihexadecyl phosphate (DCP), and levofloxacin were obtained from Sigma-Aldrich, Inc. (St. Louis, MO, USA). Other chemicals and reagents were obtained from Millipore Sigma (St. Louis, MO, USA) and were used without further treatment.

### 2.1. Ethical Approval and Declaration

The study protocol was approved by the Department of Graduate Studies, University of the West Indies, St. Augustine, Trinidad and Tobago. The application for animal research was approved by the Animal Ethics Committee, Faculty of Medical Sciences, University of the West Indies. The research was conducted using Sprague Dawley rats. Protocols for animal handling followed the Principles of Laboratory Animal Care (LAC) and the Institutional Animal Care Committee (IACC) and in accordance with the National Institutes of Health’s “Guide for the Care and Use of Laboratory Animals” (NIH publication no. 8623).

### 2.2. Experimental Design for Niosome Formulation

The input variables were entered into the Fusion One Design of Experiment (DoE) module as follows—Design wizard mode: user interactive; design type: two levels (full factorial); number of internal blocks: 1; blocking strategy: no blocking; center point level settings: CHOL, 10; SAA, 10; DCP, 3.75, and variable level settings: CHOL (mM), 5.0 < CHOL < 15.0; SAA (mM), 5 < SAA < 15; DCP (mM%), 2.5 < DCP < 15. The layout of treatment parameters is shown in Table 1.

### 2.3. Niosome Preparation and Characterization

Niosomes were prepared following the previously reported and optimized formulation comprising cholesterol, sorbitan monostearate (Span 60), and dihexadecyl phosphate (dicetylphosphate) in the molar ratio of 9.5:9.5:1, respectively, using the thin film hydration method [22,23]. Niosomes containing levofloxacin were prepared using surfactant (Span 60), cholesterol, and dicetylphosphate in varying molar ratios determined by Design of Experiment (DoE Fusion One software). The layout of factor combinations is shown in Table 1. The thin film hydration and post-preparation handling procedures, previously reported in our lab [3,22], were used. Hydrated pro-niosomes were ultra-sonicated for 6 min using a 20% amplification setting at 25 °C. The resulting colloidal dispersion was washed twice with phosphate-buffered saline and lyophilized. The lyophilized powder was stored in an air-tight, light-protective vial until used for further experimentation.

#### 2.3.1. Characterization of NiosomesScanning Electron Microscope (SEM) of Niosomes

##### Scanning Electron Microscope (SEM) of Niosomes

SEM analysis was performed at the George Washington University (GW) Nanofabrication and Imaging Center. To determine the size of the particles, all lyophilized compounds were mounted on aluminum stubs using carbon tape for SEM analysis. Imaging was performed using an in-lens FEI Teneo FEG SEM (Thermo Fisher, Allentown, PA, USA). High-resolution images were acquired under high-vacuum conditions, with a voltage of 2 kV, a beam landing current of 25 pA, and a working distance of 3.9 mm. The horizontal field width was set to 11.8 µm, and the pixel size was 1.93 nm. A dwell time of 1 µs was used, along with a resolution of 6000 × 6000 pixels for each SEM image. No coating was applied during these analyses.

##### Transmission Electron Microscopy

A total of 5 mL of nanoparticle suspension was applied for 5 min onto formvar/carbon-coated 200 mesh copper grids (Electron Microscopy Sciences, Hatfield, PA, USA) that were glow discharged (20 mA for 60 s) right before use. EVs were negatively stained with 1% uranyl acetate (Electron Microscopy Sciences) for 2 min and air-dried before imaging. A FEI Talos F200X transmission electron microscope (Electron Microscopy Sciences, Hatfield, PA, USA) operated at 200 kV was used. Images were acquired with a Thermo Scientific Ceta 16M CMOS camera (Thermo Fisher Scientific, Washington, DC, USA).

##### Vesicle Size, Polydispersity Index (PDI), and Zeta Potential

The hydrodynamic size, size distribution, and polydispersity index (PDI) of the niosomes were determined using a 90Plus particle Size Analyzer (Brookhaven Instruments, NY, USA). About 100 µL of freshly prepared niosomes was diluted to 2 mL in deionized water, loaded into culverts, and analyzed following the manufacturer’s protocols. The surface charge (Zeta potential) was also determined using the 90Plus particle Size Analyzer.

### 2.3.2. Assay of Niosome Drug Content

Amounts of drugs in starting, intermediate, and finished niosomes were analyzed using validated methods on an Agilent 1260 Infinity HPLC machine.

#### HPLC Method Development

The HPLC system comprised the Agilent 1260 Infinity II LC Gradient DAD System, incorporating a gradient pump with degasser (max. pressure 600 bar), autosampler, and column oven and equipped with OpenLab CDS Workstation PC Bundle (Agilent Technologies, Savage, MD, USA). The column was Poroshell 120 EC-C18 4.6 × 100 mm, 2.7 μm. The mobile phase comprised acetonitrile: water, each with 0.75% trifluoro acetic acid (TFA) operated in gradient mode. The injection volume was 20 μL, and the column temperature was at 35 °C. The system was validated for specificity, repeatability, accuracy, and precision following USP specifications.

The drug contents of formulated niosomes (encapsulation efficiency, EE) were calculated according to the following formula:EE = (W Encapsulated/W total) × 100%

#### In Vitro Dissolution of Drug from Niosomes

The dialysis tube method previously reported [22] was used with modification to determine drug dissolution from lyophilized niosome powder. An amount of niosomes equivalent to 5 mg of the drug was placed in a dialysis tube. The tube was suspended in an external reservoir containing 50 mL of phosphate buffer saline (pH 7.4) maintained at 37 ± 1 °C and subjected to continuous oscillatory stirring motion at 50 rpm. Aliquots of 2 mL were withdrawn at specified time intervals and replaced with fresh medium equilibrated to 37 °C. The amount of drug in each aliquot was determined by the validated HPLC method.

### 2.4. Design of Bioavailability Study

The principle of 3Rs (Reduction, Refinement, and Replacement) [24] guided the Animal Ethics Committee in approving twelve rats for the study protocol. A non-crossover, one-phase parallel experimental design was used. Sprague Dawley rats 6–8 weeks old, weighing between 200 and 250 g, were obtained from the School of Veterinary Medicine, University of the West Indies. The animals were randomly assigned to 2 groups (n = 6) and were allowed to acclimatize to the local conditions for seven days before any experimental work was performed. Group 1 animals were given the conventional (pure) drug, whilst group 2 animals were given an equivalent dose of the drug entrapped in niosomes. Levofloxacin (>98% pure HPLC, Sigma Aldrich, Oakville, ON, Canada) solution in phosphate-buffered saline (PBS) was administered intraperitoneally (i.p.) in the left lower quadrant at a dose of 7.5 mg/kg, as previously reported [9], using a 23gauge needle. The tail was cleaned with chlorhexidine gluconate, followed by an alcohol prep pack, and allowed to dry whilst the animal was restrained. The tail was snipped approximately 1–2 mm above the tip with a pair of sterile scissors and was massaged gently from top to bottom whilst blood was collected in sterile Eppendorf tubes. Blood was collected at time points 0, ½, 1, 2, 4, 6, 8, 12, 24, 48, and 72 hours after the drug was administered. The total volume of blood collected during a 24 h period was limited to ≤7.5% of the total circulating volume (estimated average of 64 mL/kg), in accordance with the NIH Animal Research Advisory Committee (ARAC) Guidelines (National Institutes of Health (US) [25].

### 2.5. Sample Preparation for HPLC Analysis

Eppendorf tubes containing blood samples were allowed to stand for 30 min. Blood samples were then centrifuged at 5000 rpm for 10 min. The serum was then carefully siphoned using a micropipette. A micropipette was used to transfer 40 µL of serum to a clean, dry Eppendorf tube (1.5 mL), followed by 120 µL of methanol to precipitate protein content. The resulting mixture was vortexed for one minute and then centrifuged for 10 min at 10,000 rpm. The supernatant was collected and stored in clean, dry Eppendorf tubes at −80 °C until used for analysis.

To obtain blank serum from rat blood for HPLC analysis, rats were euthanized using 150 mg/kg i.p. pentobarbital injection [26]. The rodent was placed lying on its back and checked for pedal reflex to ensure the animal could not feel pain. Blood was drawn by cardiac puncture. With the bevel facing upwards, the needle was inserted to the left of the animal’s xiphoid process (base of the sternum), directing the needle towards the chin at a 30–45° angle. After the needle tip penetrated the skin, the syringe plunger was gently pulled back to create slight negative pressure. The needle was advanced until blood was withdrawn. If unable to withdraw blood, the needle was slowly withdrawn while maintaining the negative pressure on the syringe until the needle was nearly but not completely withdrawn. The needle was then redirected in a slightly different direction. Blood was placed in a sterile tube and allowed to stand for 30 min. Blood samples were then centrifuged at 5000 rpm for 10 min. The supernatant was removed and placed in sterile glass tubes. Methanol was added to serum in a ratio of 3:1 for protein precipitation. The resulting mixture was vortexed for one minute and centrifuged for 10 min at 10,000 rpm. The supernatant was collected and stored in clean, dry Eppendorf tubes at −80 °C until used for analysis. To prepare the sample for analysis, 20 µL of serum was placed in a clean, dry Eppendorf tube using a micropipette. Forty microliters (40 µL) of methanol was added. The contents were vortexed for one minute and then centrifuged at 10,000 rpm for 10 min. The supernatant was removed and loaded in an HPLC autosampler tray, from which 20 µL was injected during the analysis. HPLC analytical parameters were as described under Section 2.3.2. above.

### 2.6. Treatment and PK Analysis of Plasma Data

HPLC data were entered into Gastroplus PK^TM^ (Simulations Plus, Inc., Lancaster, CA, USA) modeling and simulation software. The data obtained from the pharmacokinetic analysis of serum samples represented the drug concentrations in plasma (Cp) at various time points. The Cp vs. time data were entered in the GastroPlus™ menu, and the non-compartment analysis (NCA) was conducted.

The individual weight of each animal was then entered into the software, and pharmacokinetic modeling was determined for one, two, and three compartmental models. The software used the Hooke and Jeeves Pattern search [27,28] with the error weighting set at 1/Yhat^2^. The software was able to generate solutions for non-compartment analysis (NCA) and one-, two-, and three-compartment models. This data analysis incorporated model fitness algorithms of the Akaike information criterion (AIC) [28,29,30] and Schwarz criterion (SC) [31] according to the following equations:AIC = (#Pts) × Log(Obj) + 2(#Parameters)(1)SC = (#Pts) × Log(Obj) + (#Parameters) × (Log(#Pts))(2)

The best model was reported by each algorithm together with estimates of errors (residuals).

## 3. Results

### 3.1. Niosome Physicochemical Properties and Encapsulation Efficiency

The SEM images of levofloxacin niosomes are shown in Figure 1A,B. Encapsulation efficiency (EE) ranged from 9.5% (Formulation #6) to 34.2% (Formulation #11), PDI ranged from 0.216 (Formulation 1) to 0.392 (Formulation 12) while vesicle size ranged from 203 nm (Formulation #1) to 783.1 (Formulation #10). The goal of a good nanoencapsulation delivery system is to minimize vesicle size (for stability and cellular internalization via biological membranes) and to maximize EE and polydispersity index. Also, a positive Zeta potential has been suggested but is still subject to controversy. Table 2 shows that the regression of vesicle size on vesicle-forming input variables (CHOL and Span 60) was not statistically significant (*p* = 0.066). This indicates that other variables, including charge inducer and drug molecules, might be influencing vesicle formation. Table 3 shows the model ranking of input variables in the order of SP60 > CHOL > SP60-CHOL interaction). Data from the screening experiments were inputted into the DoE Optimizer module. The predicted optimized formulation for achieving particle size minimization was prepared in triplicate, and the particle size was determined. Model verification was performed by comparing the vesicle sizes of the suggested optimized solution obtained from laboratory data to the predicted vesicle size (Table 4). The average vesicle size and PDI of the experimental formulation were 330.92 (±177.19) and 0.273 (±0.06), respectively, compared to the predicted values of 329.16 nm and 0.286. The difference between experimental and predicted vesicle size and PDI, respectively, were not statistically significant (*p* > 0.05). The observed mean encapsulation efficiency (EE) was 19.56 (±9.14). Preparation of test niosomes using DoE Optimizer recommended factors combination produced niosomes with an average size of 329.16 nm ± 18.0 (SE), PDI of 0.286 ± 0.014 (SE), EE of 30.74%, and Zeta potential of 21.72 ± 0.54 (SE), each of which fell within the predicted (−/+)2 Sigma range of encapsulation efficiency (Table 4). This was used in animal studies.

Typical plots of size distribution and Zeta potential of niosomes are shown in Figure 2A,B.

The impact of input variables on niosome characteristics is predicted by the Equations (3) to (5):Vesicle size (nm) = + 322.595 + 73.703(CHOL) − 76.230(CHOL × SP60) + 97.330(SP60 × DCP)(3)EE (%) = + 27.428 + 4.318 (SP60 × X3)(4)Zeta = −20.724 + 19.180(SP60) + 34.761(SP60) + 32.920(CHOL × SP60)(5)

The interrelationship of niosome formers on vesicle size, encapsulation efficiency, and Zeta potential are shown in the response surface graphs in Figure 3A, B, and C, respectively.

The response surface graphs of niosome properties and input variables are shown in Figure 3B,C. Generally, increasing the ratio of cholesterol to Span 60 causes a significant decrease in the average particle size.

### 3.2. HPLC Analytical Method Parameters

The calibration samples of levofloxacin at different concentrations were prepared and analyzed on HPLC using parameters described in Section 2.3.2. The chromatogram obtained using a mobile phase of water: acetonitrile (+0.75% trifluoro acetic acid, TFA) in gradient mode sowed levofloxacin elution at an average retention time of 5.8 (±0.047) minutes (Figure 4A). The peak area obtained for each sample was plotted against the respective concentration (Figure 4B). The equation of the line (Equation (1)) obtained from the calibration curve was as follows:Peak Area = 294.06 × Conc − 54.583(6)

The correlation between peak area versus concentration plot was near perfect linearity with R² = 0.9987. The parameters of Eqn. 6 were used to convert areas of chromatograms obtained for the analytes to drug concentrations and independently confirmed with intermittent analysis of known concentrations of reference samples interspersed between each set of analytes run on HPLC. The HPLC analytical parameters are shown in Table 5. The dependence of the chromatogram’s AUC (mAU/s) on analyte concentration in assayed samples is reflected by the very high regression coefficient (R^2^ > 0.99). The validity of the analytical method is supported by the high accuracy of 99.7% (USP range: 85–155%) and mean recovery of 97.9% (USP range: 85–115%). The stability of the analytical method is shown by the intra-day and inter-day data shown in Figure 5. The intra-day and inter-day validation runs on HPLC showed variations of less than 6.12% and reproducibility with a relative error of less than 2.5% (Table 6). The upper and lower limits (Figure 5) are within 95% and 110%.

### 3.3. Drug Dissolution from Niosomes

The dissolution profiles of levofloxacin from screening experiments and optimized niosome formulation with equivalent encapsulation efficiencies are shown in Figure 6. Optimized formulation showed a slightly higher release rate until about 5 h. Less than 80% was released in 6 h, while about 100% was released in 8 h from both formulations. Niosome encapsulation effectively protected the drug from rapid release, typical of the pure drug from conventional dosage forms. Niosome formulation could extend the short half-life of pure levofloxacin.

### 3.4. Non-Compartmental Analysis (NCA) in GastroPlus™

Results of data treatment in the PK module of the Gastroplus modeling and simulation software indicated the weighted sum of squared errors = 7.7402 × 10^−1^ and a weighting of 1/Yhat^2^. Figure 7A–C display the results of levofloxacin concentration versus time (Cp vs. t), log-transformed Cp vs. t, and the plots of residuals, respectively, for levofloxacin formulation in niosomes. Similarly, Figure 8A–C display the respective results of pure drug levofloxacin following intraperitoneal administration in rats. Plots in Figure 8A,B show discernible bi-exponential profiles of the respective Cp versus t and log Cp versus t plots, which are reflective of the two-compartment open model. The comparative bioavailability profiles of levofloxacin and its niosomes are shown in Figure 9.

Non-compartmental PK parameters are useful for predicting biodisposition patterns of drug molecules from delivery systems into and outside of the biological system. The area under the concentration versus time plot (AUC) is an indication of the rate and extent of drug exposure to the body. AUMC is the area under the first moment curve. It is obtained as the product of concentration and the time it was obtained. When AUMC is divided by the AUC, the mean residence time (MRT) is obtained. MRT is an index of how long the drug molecule stays at different sites in the body. As shown in Table 7, AUC, AUMC, and MRT values are higher for niosomes than for levofloxacin. The comparative values of MRT is also shown in Figure 10. Conversely, the rate of removal of drugs from the body (clearance, Cl) is slower for niosomes than for the pure drug, indicating the ability of niosomes to sustain the intravascular residence of the embedded drug.

The algorithmic search in Gastroplus for the most fitting PK model (using Hooke and Jeeves patterns search [27] and objective weighting of 1/Yhat^2^) indicated that both Akaike information criterion (AIC) [29] and Schwartz criterion (SC) [30,31] support the one-compartment as the preferred model for PK of drug in niosomes and two-compartment as the preferred model for the PK of pure levofloxacin.

## 4. Discussion

### 4.1. Niosome Formulation and Characterization

Span 60 (SP60) was selected as a niosome-forming, non-ionic, lipid-soluble surfactant, cholesterol (CHOL) was used as a membrane stabilizer, and dihexadecyl phosphate (DCP) was used as a charge inducer for the prepared niosomes based on our previous studies [3,22]). Using a 2^3^ factorial experiment with four center points (total formulation = 12), levels of SP60, CHOL, and DCP for the control niosomes were screened. Input data from the screening experiment (Span 60/CHOL/DCP) and the corresponding output variables were inputted into the Optimizer module, and the predicted responses generated are shown in Table 4. The Optimizer recommended variable molar levels (Span 60/CHOL/DCP, 15:24:9) were used to prepare the niosomes used in the BA study.

The size, PDI, and Zeta potential of niosomes are determined by the formulation ingredients, and the media in the niosomes are suspended [32]. Zeta potential and particle size affect the thermodynamic properties of the niosomes. Because the smaller-sized niosomes have a larger surface-to-volume ratio, they tend to be more thermodynamically stable. The particle size distribution and polydispersity index (PDI) of niosomes can affect the processability, bulk properties, product appearance, and biological performance of the product [32]. Thus, reliable and reproducible analytical procedures for niosomes’ mean vesicle diameter, surface charge, and heterogeneity are important fundamental quality control product parameters. The small niosome sizes and the high value of Zeta potential imply good stability of the colloidal dispersion, while the reasonably high %EE indicates the ability to deliver therapeutic concentration of niosomes intravascularly via i.p. injections. Cholesterol can enhance the bilayer hydrophobicity by its aliphatic chain aligning parallel to the hydrocarbon chains of the amphiphilic surfactant (Span 60) and its hydroxyl group of the sterol moiety forming a hydrogen bond with the ester group in Span 60. This alignment will cause a decrease in surface free energy, thereby promoting vesicle size reduction. Thus, vesicle size increases with the proportion of Span 60 and decreases with that of cholesterol (Figure 3A), while encapsulation efficiency increases with the proportion of Span 60 and decreases with cholesterol for hydrophilic drugs like levofloxacin (Figure 3B).

### 4.2. PK Modeling and Analysis

From the data generated for the NCA in Table 6, the AUC, AUMC, MRT, Cl, and Vss are all above the 125% range for bioequivalence. The coefficient of variation (CV) accompanying various parameter values of the compartmental models in Table 7, Table 8 and Table 9 is the measure of errors or residuals for which the respective models cannot account. Thus, the higher the CV values, the less powerful the model is in representing the PK profiles for the biodisposition of the drug. As shown in Table 7, Table 8 and Table 9, CV values for a given parameter increase with an increase in the number of the model’s compartments. On the other hand, R^2^ is a measure of the model’s strength correlating log plasma concentration with time. The higher the value (i.e., the closer it approached perfect linearity with R^2^ = 1), the stronger and more predictive the model. For the one-compartment model, R^2^ = 0.418 for niosomes and R^2^ = 0.335 for unencapsulated drugs. When escalated to the two-compartment model, however, R^2^ = 0.418 for niosomes, while R^2^ = 0.5749 for pure drug. Unlike the simple linear models, the Gastroplus platform has a more integrative model search capability involving the use of powerful algorithms of AIC and SC criteria with appropriate error weighting.

The summary Cp versus t plots of niosomes and non-encapsulate drugs are shown in Figure 9, while their corresponding AUC of niosomes and pure levofloxacin are shown in Table 7, Table 8 and Table 9. Bioequivalence of a given product is conferred if the mean bioavailability of the test product under consideration is within 80–125% of the reference formulation. From the perspective of pharmacokinetic modeling, the FDA 2003 guidance indicated that other parameters, such as AUC_(0–∞)_, AUC_(0–tlast)_, and C_max_, should also be considered [33]. The values of AUC, AUMC, and MRT were all significantly higher for the niosomal formulation than for the unentrapped drug (Table 7 and Table 8). The decreased volume of distribution, decreased clearance, and enhanced MRT and AUC are positive outcomes of niosomal encapsulation, which is indicative of the potential of niosomes to increase the drug’s therapeutic efficacy whilst concurrently minimizing its adverse effects. While the one-compartment behavior of levofloxacin niosomes and the corresponding increase of MRT and reduced CL and Vss cannot be easily explained, it appears that the niosomal formulation prevented rapid drug release and released it in a sustained manner. Therefore, the relative bioavailability and the MRT of the niosome-encapsulated drug were increased above those of unencapsulated drug molecules.

Fluoroquinolones generally have a superior ability to penetrate tissues [34] than other antibiotics and can penetrate organelles like macrophages and neutrophils to accentuate bactericidal activity [35,36,37]. Analysis of the data generated for the one-compartment model of levofloxacin shows several parameter differences between the niosome-encapsulated formulations and the conventional pure drug. All parameters tested were out of the bioequivalence range, indicating significant differences in their pharmacokinetic parameters. Differences in all parameters tested were significant at a 95% confidence interval, except for MRT, Vd, and t_1/2._ Even though the MRT, Vd, and t_1/2_ showed no significant difference, the MRT for the niosome formulation (40.62 h) was 1.64 times higher than that of the unentrapped conventional formulation (24.66 h). The same was observed for the t_1/2_ of the niosome-entrapped drug (28.66 h), which shows a half-life of 1.64 X than that of the unentrapped drug (17.09 h). Thus, the niosome-entrapped drug showed a higher effect on all other parameters than the unentrapped drug. The only possible reason for this difference was the drug delivery system employed. Encapsulation in niosomes seems to restrict drug distribution, enhance plasma levels, and allow for sustained release of drug molecules from the construct. In the two-compartment model (Table 9), only the values of Vc and C_max_ were bioequivalent, with all other parameters falling outside of the stipulated 80–125% range. The data, however, were not significantly different, as can be seen from the “t” values, as well as the p values generated for these data (Table 9). Progression to the three compartmental analysis (Table 10) resulted in PK parameters losing reliability as the coefficient of variation increased dramatically.

The Hooke and Jeeves pattern search [27] also showed a difference in the release and pharmacokinetic characteristics of the two formulations. The patterns shown in Figure 7A,B indicate that niosome formulation shows a linear pattern of drug release. The initial post-absorption rate of drug release from the niosomal formulation was much slower and more sustained than those from the conventional free drug (Figure 8A,B). The logarithmic display for niosome (Figure 7B) and conventional drug (Figure 8B) show a clear difference in release pattern as the niosome profile is linear compared to the nonlinear profile of the conventional drug. The plot of residuals for the niosome-treated animals (Figure 7C) showed an even distribution around the horizontal axis that supports a linear profile of drug release. This even distribution was not seen in the plot for the conventional formulation (Figure 8C). The average Cp vs. time profiles for rats administered levofloxacin niosomes and those with pure levofloxacin are shown in Figure 9. Similar findings have been reported by other studies in the literature. Ruckmani et al. (2010) [38] and Ammar et al. (2017) [39] showed that niosome entrapment of the drugs enhanced the AUC, MRT, and t_1/2_. Feitosa et al. (2022) [40] developed a niosomal formulation of doxycycline using Span 60/Tween 60 and cholesterol via the modified thin hydration method. The average size of the final formulation was 281.9 nm, and encapsulation efficiency (EE) was 72.1%. A significantly lower minimum inhibitory concentration (MIC) against different Gram-positive and Gram-negative bacteria was reported, indicating higher antibacterial activity of doxycycline niosomes than that of the free drug [4,40].

To determine the model that best fit the drug exposure in the rats, the model search was extended from non-compartmental (purely mechanistic) to physiologic models depicting the body compartments into one, two, or three, depending on the rate of perfusion and rapidity of equilibration between plasma (central compartment) and various organs (rapidly equilibrating organs and tissues) [41,42]. While noting that PK models cannot indicate the absolute physiological or anatomical location of drug molecules as they traverse the body of an animal [43], a close approximation of one model relative to another may provide the most accurate representation of drug exposure systemically. Presently, many drugs need to be administered at sufficient doses to obtain the required dose at the desired site of action, which may often be outside of the plasma compartment. The PK parameters selected based on the most predictive model could be advantageous in the choice of delivery design that would provide the most efficient drug dosing. For instance, as was previously reported [22], a 50% reduction in MIC/MBC of some fluoroquinolone niosomes (including levofloxacin, gatifloxacin, and ciprofloxacin) on drug-resistant strains of *P. aeruginosa*, *E. coli*, and *S. aureus* could enable dose reduction if the PK parameters indicate that drug perfusion is actually enhanced by niosome encapsulation of the drug molecules. Therefore, the slower rate of distribution of niosome-encapsulated levofloxacin from plasma to the extravascular compartment (niosomes kz = 0.025; pure drug kz = 0.041) and the significantly higher mean residence time (MRT, Figure 10) suggest a high potential for the translation of the observed reduction in MIC/MBC in vitro to in vivo microenvironment.

Encapsulation in sub-microscopic particles generally has been associated with improved bioavailability. Until recently, liposomes have been used extensively to improve the pharmacokinetic profile of drugs for administration via different routes [44,45]. The results of those studies indicate the ability of the drug carrier to preserve the drug en route distribution channels to the plasma, protect it from the external environment, and preferentially deliver it to the site where its action is needed. These outcomes have inspired the FDA approval of some liposomal formulations for clinical use. However, the short in vivo fate of liposomes is a limiting factor, and studies have focused on improving its circulation time [44]. Unlike liposomes, niosomes are cheaper to produce and do not require special storage and handling conditions. Hence, recent attention has been focused on niosomal drug delivery, as in this study. Taken together with our previous report [22], this PK model analysis indicates that niosome encapsulation of levofloxacin by concentrating the drug in the intravascular compartment, preventing its binding to plasma protein, and preventing bacteria sensing of drug molecules that are enveloped in lipid vesicles (niosomes) has the potential to obviate resistance development, which is currently limiting the efficacy of the drug.

## 5. Conclusions

Niosome formulation containing Span 60, cholesterol, and dihexadecyl phosphate at a 15:24:9 molar ratio produced encapsulation efficiency (EE) of 32.47%, mean hydrodynamic vesicle size of 320.19 nm (±18.0 nm), PDI of 0.316 ± 0.014 (SE), and Zeta potential of 21.72 ± 0.54 (SE), each of which fell within the predicted (−/+)2 Sigma range. This optimized levofloxacin-loaded formulation was used for the BA studies in rats. All PK parameters tested were out of the bioequivalence range, indicating significant differences in pharmacokinetic parameters of nio-encapsulated and non-encapsulated levofloxacin. Differences in all parameters tested were significant at a 95% confidence interval, except for Vd and t_1/2._ The MRT for the niosome formulation (40.62 h) was 1.64 times higher than that of the unentrapped conventional formulation (24.66 h). The same was observed for the t_1/2_ of the niosome-entrapped drug (28.66 h), which shows a half-life of 1.64 x than that of the unentrapped drug (17.09 h). Niosomes appear to protect levofloxacin en route into plasma, altering its pharmacokinetic profile by preventing its binding to plasma protein and increasing its residence time in the systemic circulation. This suggests significant potential for niosomal delivery of levofloxacin to intravascular bacteria. Taken together with our previous report, this PK model analysis indicates that niosome encapsulation of levofloxacin could prevent bacteria sensing of the drug molecules, thereby obviating resistance development, which is currently limiting the efficacy of the drug. Further studies are needed to confirm intracellular delivery and the delivery mechanisms of niosome-encapsulated levofloxacin into multi-drug-resistant bacterial cells.

## Figures and Tables

**Figure 1 pharmaceutics-17-00275-f001:**
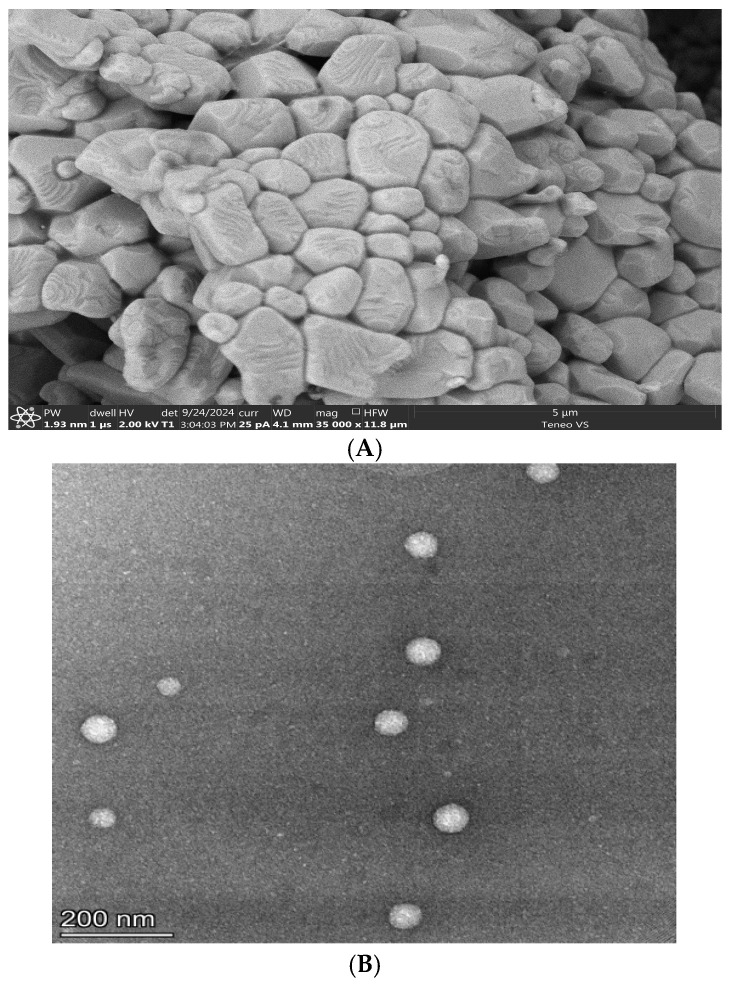
(**A**) Scanning electron microscope of levofloxacin niosomes. (**B**) Transmission electron microscopy of levofloxacin niosomes.

**Figure 2 pharmaceutics-17-00275-f002:**
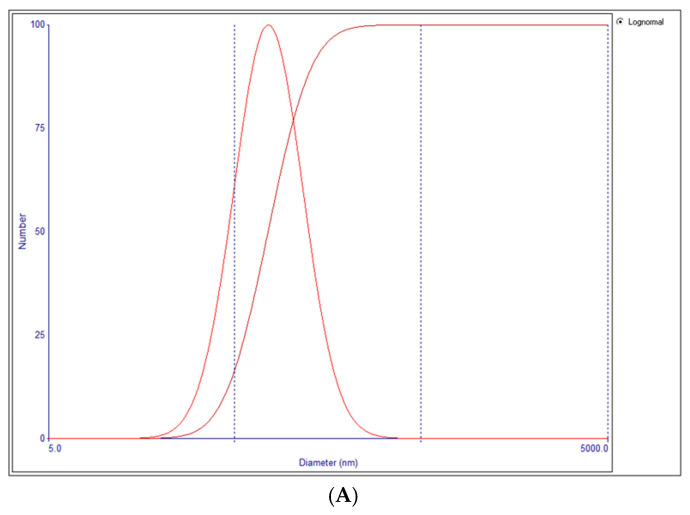
(**A**) Typical niosome size distribution. (**B**) Typical niosome Zeta potential plot.

**Figure 3 pharmaceutics-17-00275-f003:**
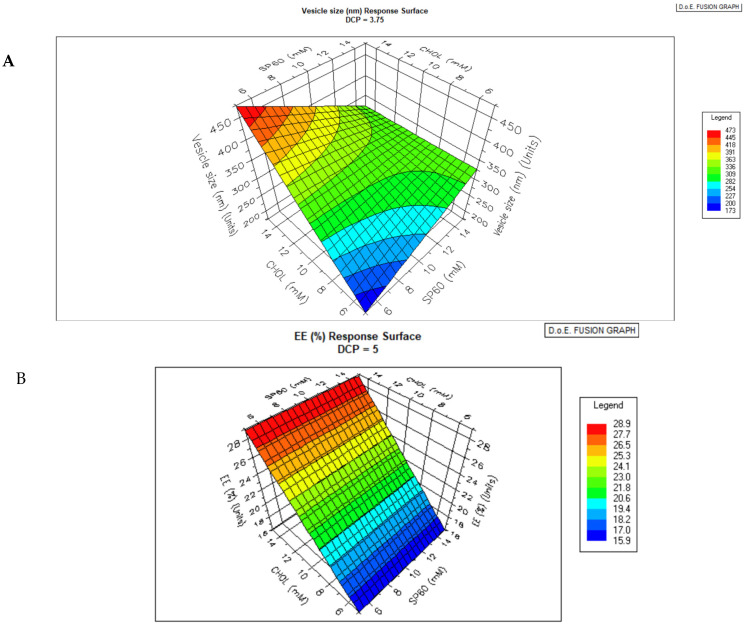
(**A**) Response surface for the interaction between input variables and their impact on (**A**) niosome drug encapsulation. (**B**) Response surface for the interaction between input variables and their impact on vesicle (niosome) size. (**C**) Response surface for the interaction between input variables and their impact on polydispersity index (PDI). Legend represents vesicle size, EE or PDI respectively.

**Figure 4 pharmaceutics-17-00275-f004:**
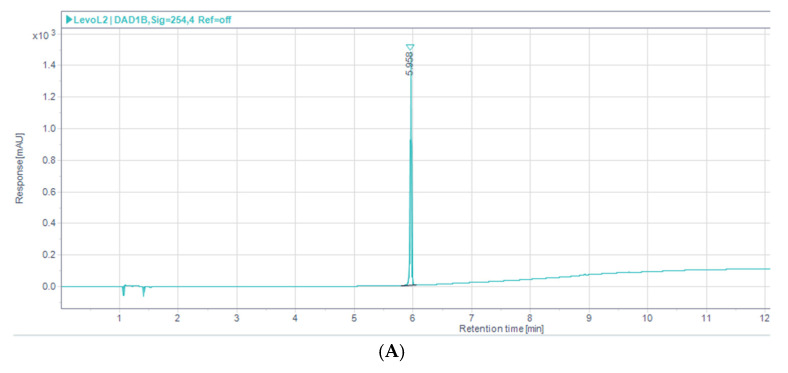
(**A**) Typical chromatogram of levofloxacin on reverse phase HPLC column (ACN:H_2_O, +0.75%TFA gradient mode). (**B**) Reverse phase HPLC calibration curve for levofloxacin in the method (ACN:H_2_O; 60:40). Legend represents vesicle size, EE or PDI respectively.

**Figure 5 pharmaceutics-17-00275-f005:**
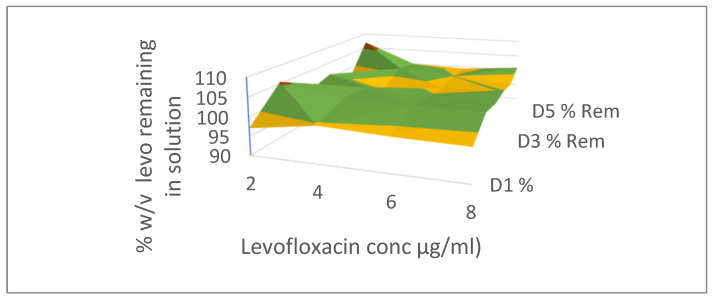
Intra-day and inter-day stability of levofloxacin concentrations during HPLC analysis.

**Figure 6 pharmaceutics-17-00275-f006:**
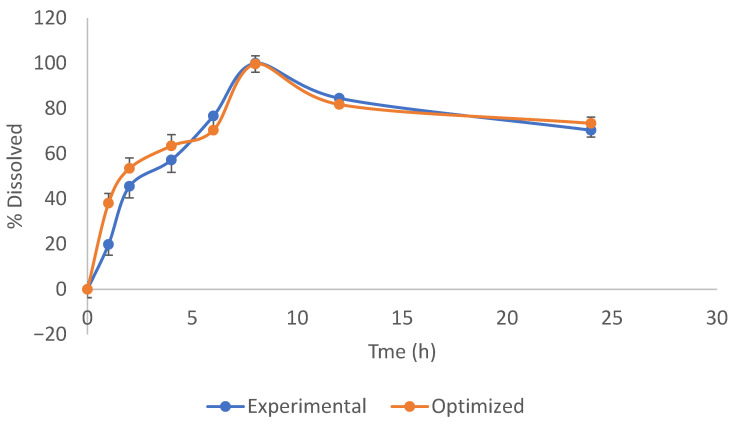
Dissolution profiles of levofloxacin-loaded niosomes from screening (red) and optimized (blue) formulations.

**Figure 7 pharmaceutics-17-00275-f007:**
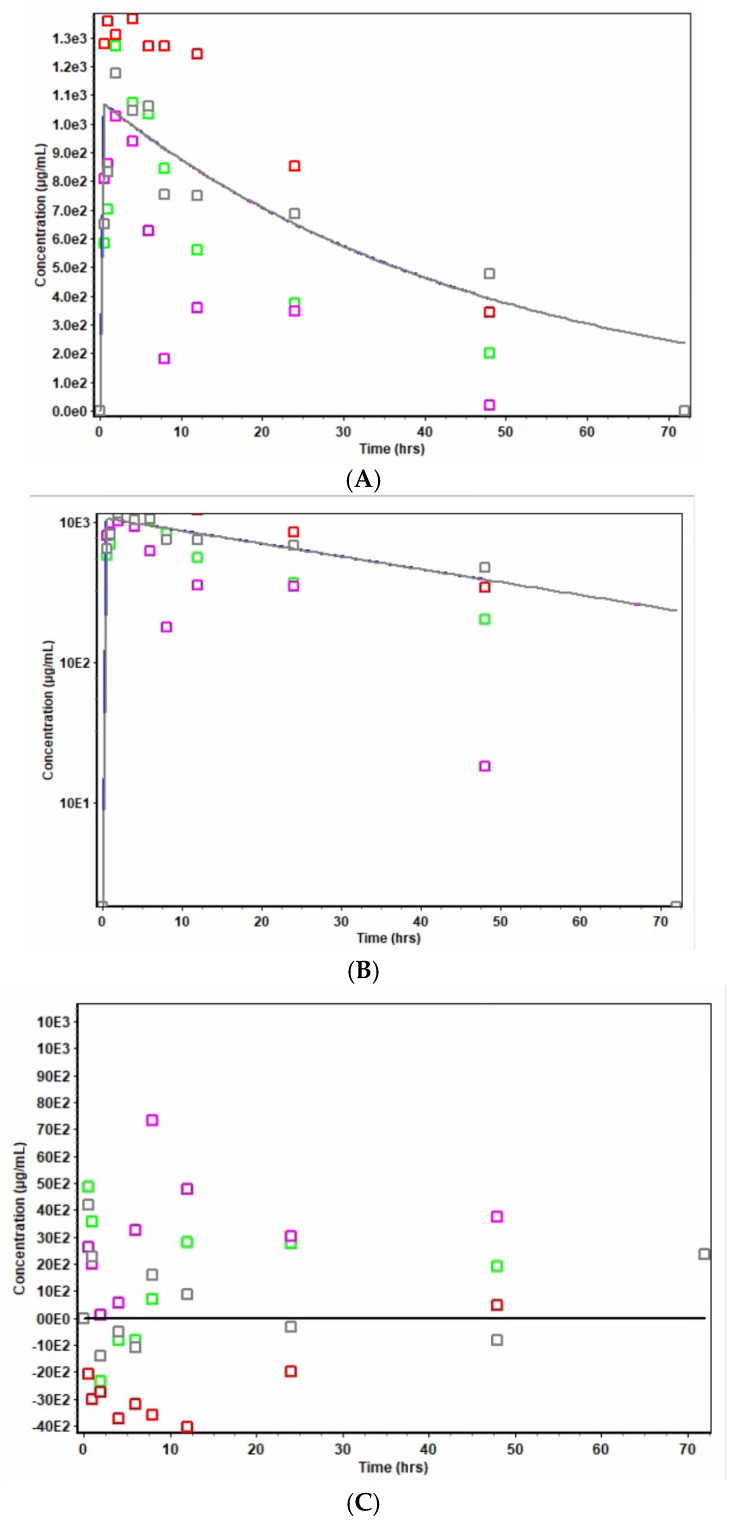
(**A**) Cp vs. t for levofloxacin niosomes. Each colored box represents a Cp vs. time profile of an animal; lines represent lines of best fit. (**B**) log Cp vs. t for levofloxacin niosomes. Each colored box represents a Cp vs. time profile of an animal; lines represent lines of best fit. (**C**) Plot of residuals for levofloxacin niosomes. Each colored box represents residuals for a Cp vs. time profile of an animal; lines represent lines of best fit.

**Figure 8 pharmaceutics-17-00275-f008:**
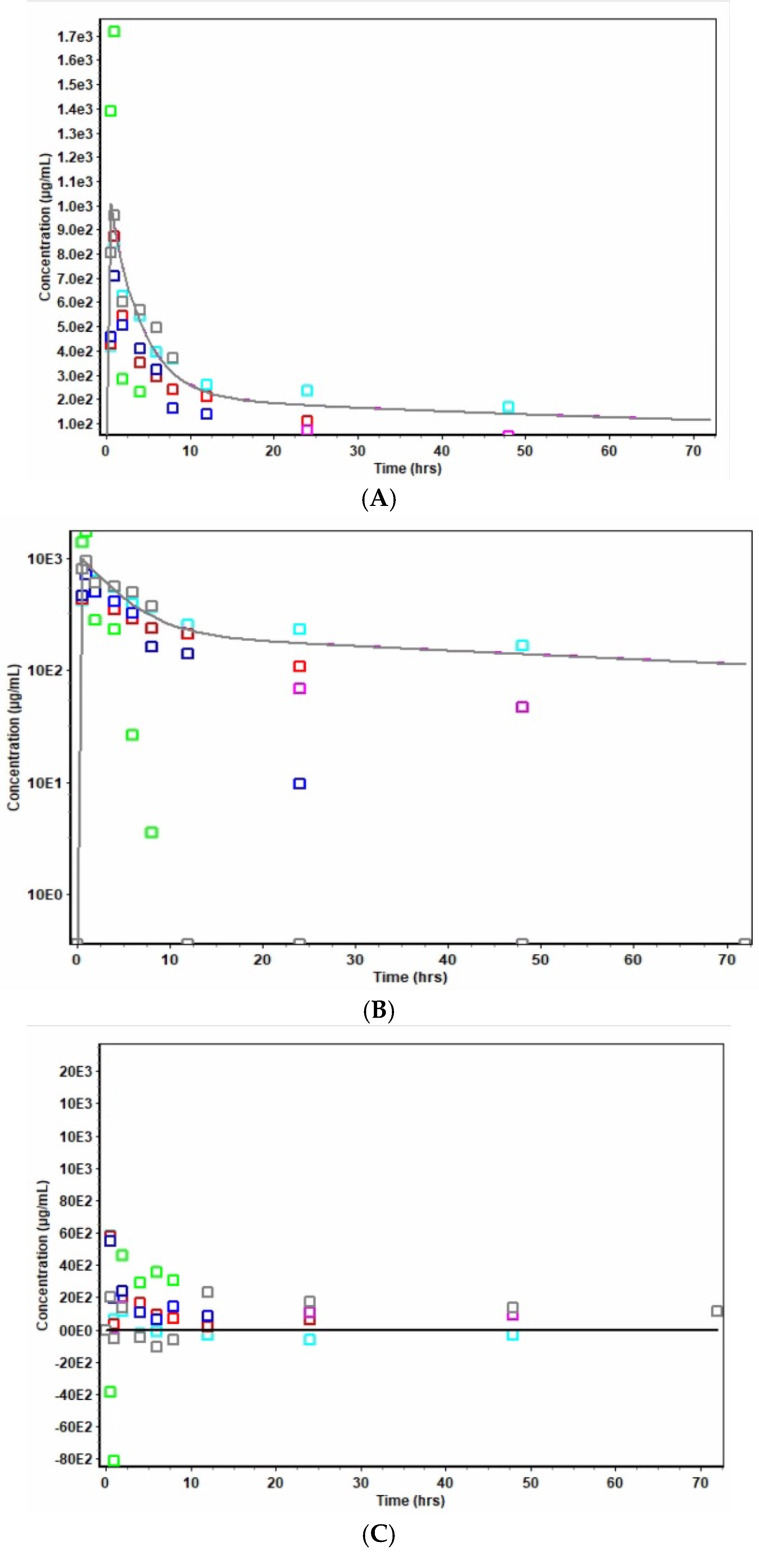
(**A**) Cp vs. t for pure unencapsulated levofloxacin. Each colored box represents a Cp vs. time profile of an animal; lines represent lines of best fit. (**B**) log Cp vs. t for pure unencapsulated levofloxacin. Each colored box represents a Cp vs. time profile of an animal; lines represent lines of best fit. (**C**) Plot of residuals for pure, unencapsulated levofloxacin. Each colored box represents residuals for a Cp vs. time profile of an animal; lines represent lines of best fit.

**Figure 9 pharmaceutics-17-00275-f009:**
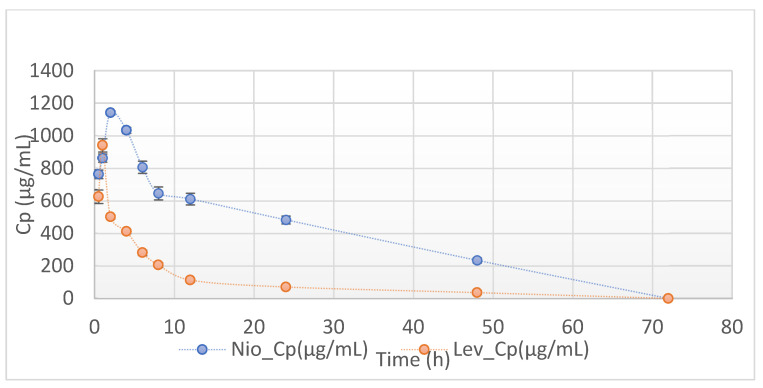
Average Cp vs. times for rats administered levofloxacin niosomes and those with pure levofloxacin.

**Figure 10 pharmaceutics-17-00275-f010:**
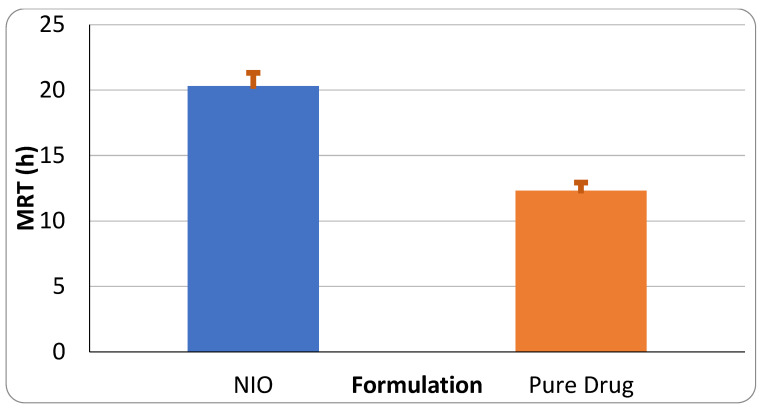
Mean residence time (MRT) of niosomes vs. pure drug levofloxacin.

**Table 1 pharmaceutics-17-00275-t001:** Excipient molar ratio (input variables) and response variables from screening experiments.

Run No.	CHOL (mM)	SP60 (mM)	DCP (mM%)	EE (%)	PDI	* Vesicle Size (nm)	Zeta (mV)
1	15	5	5	24.921	0.216	203.1	0.7
2	5	15	2.5	29.059	0.230	238.8	−50.83
3	15	15	2.5	30.118	0.234	249.7	−0.35
4	5	5	2.5	22.230	0.239	212.9	0.65
5	5	15	2.5	20.841	0.237	261.7	−79.67
6	5	5	5	9.549	0.234	204.2	0
7	5	15	5	22.8169	0.271	355.9	0
8	5	5	2.5	8.051	0.281	248.5	−48.71
9	10	10	3.75	10.229	0.247	265.3	−42.1
10	15	5	2.5	10.482	0.380	783.1	−125.45
11	15	15	5	34.213	0.316	368.8	−147.36
12	15	5	2.5	12.39788	0.392	579	−125.45

CHOL = cholesterol; SP60 = Span 60 (surfactant, SAA); DCP = dihexadecyl phosphate; EE (%) = encapsulation efficiency; PDI = polydispersity index. * Vesicle size (nm) is the hydrodynamic diameter.

**Table 2 pharmaceutics-17-00275-t002:** Regression ANOVA statistics for vesicle size.

Source of Variation	Sum of Squares	Degrees of Freedom	Mean Square	F-Ratio	*p*-Value
Regression	102,883.119	1	102,883.119	4.24	0.066
Residual	242,484.078	10	24,248.408		0.903
Lack-Of-Fit	17,860.654	3	5953.551	0.19	
Pure Error	224,623.424	7	32,089.061		
Total	345,367.197	11			

**Table 3 pharmaceutics-17-00275-t003:** Model term ranking for encapsulation efficiency.

Model Term Name	Model Term Range	Coefficient Value	Model Term Effect	Model Term Rank
X2	2.000	11.757	23.513	1.000
(X1)^2^	1.000	21.970	21.970	0.934
X1 × X2	2.000	−9.915	−19.829	0.843

X1 = CHOL; X2 = SP60.

**Table 4 pharmaceutics-17-00275-t004:** Independent variable setting and predicted response data from the DoE Optimizer.

Response Variable Name	Relative Importance	Target	Optimizer Answer	−2 Sigma	+2 Sigma
			Predicted Response	Confidence Limit	Confidence Limit
EE (%)	1	Maximize	30.738	13.381	48.095
PDI	1	Maximize	0.286	0.245	0.328
Vesicle Size (nm)	1	Minimize	329.160	41.103	617.217
Zeta	1	Maximize	−51.609	−124.639	21.421

Predicted response data: desirability target = 1.0; desirability result = 0.6.

**Table 5 pharmaceutics-17-00275-t005:** HPLC analytical parameters.

Analytical Parameters	Results	USP Limits
Linearity	y = 294.06x; R² = 0.999	R² = 0.980
Mean RT (min)	1.090 (±0.05) min	-
Accuracy (%)	99.66% (±3.36)	85–115%
LOD μg/mL	10.00	-
LOQ (μg/mL)	2.25	-
Mean Recovery	97.90% (± 1.05)	85–115%

**Table 6 pharmaceutics-17-00275-t006:** A measure of precision and accuracy in the determination of levofloxacin concentration in rat serum. (n =18).

Concentration µg/mL	Mean ± SD	RE (%)	Intra-Day Variation (%)	Inter-Day Variation (%)
2	2.025 ± 0.083	1.242	6.116	4.451
4	4.100 ± 0.091	2.503	2.341	4.419
6	6.041 ± 0.123	0.683	2.837	1.585
8	8.023 ± 0.135	0.288	2.724	2.346

**Table 7 pharmaceutics-17-00275-t007:** Non-compartment open model for levofloxacin in Sprague Dawley rats.

		Niosomes		Pure Levofloxacin	
Parameter	Units	Value	STD	Value	STD
AUC_(0-inf)_	µg-h/mL	2.78 × 10^4^	12,647.160	6.89 × 10^3^	4236.170
AUMC	µg-h^2^/mL	5.75 × 10^5^	309,275.070	9.63 × 10^4^	124,749.700
MRT	h	20.7	3.410	13.980	8.310
CL	L/h	7.12 × 10^−5^	0.000	3.0 × 10^−4^	0.000
K_(z)_	1/h	4.02 × 10^−2^	0.043	5 × 10^−3^	0.008
V_ss_	L	1.39 × 10^−3^	0.001	2.5 × 10^−3^	0.002
C_(0)_ bolus	µg/mL	7.64 × 10^2^	763.850	626.11	416.460

**Table 8 pharmaceutics-17-00275-t008:** One-compartment open model for levofloxacin in Sprague Dawley rats.

PK Parameters	Niosomes	Pure Levofloxacin		Pure Levofloxacin	
Parameters	Units	Values	CV	Values	CV
Vd	L	1.64 × 10^−3^	43.96%	2.20 × 10^−3^	0.86%
CL/kg	L/h/kg	1.81 × 10^−4^	19.16%	3.95 × 10^−4^	0.53%
Vd/kg	L/kg	7.37 × 10^−3^	43.96%	9.75 × 10^−3^	0.86%
K10	1/h	0.025	47.95%	0.041	1.01%
C_max_	µg/mL/mg Dose	665.1	-	453.9	-
t_1/2_	h	28.16	47.95%	17.09	1.01%
AUC	µg-h/mL	3.91 × 10^4^	51.64%	7327.5	1.14%
AUMC	µg-h^2^/mL	1.70 × 10^6^	51.64%	4.55 × 10^5^	1.14%
MRT	h	40.62	47.95%	24.66	1.01%
R^2^		0.418	-	0.335	-

Model parameters: Akaike information criterion (AIC) = (#Pts) × Log(Obj) + 2(#Parameters) = −11.3715; Schwarz criterion (SC) = (#Pts) × Log(Obj) + (#Parameters) × (Log(#Pts)) = −7.1828; optimization time: 7.422 × 10^−2^ s; total simulations: 594; weighted sum of squared errors = 7.740 × 10^−1^; weighting: 1/Yhat^2^.

**Table 9 pharmaceutics-17-00275-t009:** Two-compartment open model for levofloxacin and its niosome in Sprague Dawley rats.

PK Parameters	Niosomes	Pure Levofloxacin		Pure Levofloxacin	
Parameters	Units	Values	CV	Values	CV
Vc	L	1.63 × 10^−3^	53.16%	1.47 × 10^−3^	81.69%
CL_2_	L/h	1.62 × 10^−5^	6289.32%	3.08 × 10^−4^	39.17%
V_2_	L	0.034	0.00%	4.52 × 10^−3^	6.64%
CL/kg	L/h/kg	1.10 × 10^−4^	3918.40%	2.55^−4^	7.55%
Vc/kg	L/kg	7.36 × 10^−3^	53.16%	6.52 × 10^−3^	81.69%
CL_2_/kg	L/h/kg	7.28 × 10^−5^	6289.32%	1.36 × 10^−3^	39.17%
V_2_/kg	L/kg	0.152	0.00%	0.020	6.64%
A	µg/mL	1024.1	-	897.400	-
B	µg/mL	8.066	-	222.700	-
Alpha	1/h	0.025	-	0.308	-
Beta	1/h	2.85 × 10^−4^	-	8.66 × 10^−3^	-
K_10_	1/h	0.015	3918.76%	0.039	82.04%
K_12_	1/h	9.89 × 10^−3^	6289.55%	0.209	90.59%
K_21_	1/h	4.78 × 10^−4^	6289.32%	0.068	39.73%
C_max_	µg/mL	665.6	-	678.9	-
t_1/2_	h	2431.5	0.00%	80.08	0.0%
R^2^		0.4183	-	0.5749	-

Model parameters: Akaike information criterion (AIC) = (#Pts) × Log(Obj) + 2(#Parameters) = −7.369; Schwarz criterion (SC) = (#Pts) × Log(Obj) + (#Parameters) × (Log(#Pts)) = 1.009; optimization time: 0.363 s; total simulations: 2640; weighted sum of squared errors = 7.740 × 10^−1^; weighting: 1/Yhat^2^.

**Table 10 pharmaceutics-17-00275-t010:** Three-compartment open model for levofloxacin and its niosome in Sprague Dawley rats.

PK Parameters		Niosomes		Pure Levofloxacin	
Parameters	Units	Values	CV	Values	CV
CL	L/h	3.50 × 10^−5^	1407.07%	5.05 × 10^−5^	445.91%
Vc	L	1.54 × 10^−3^	1467.27%	1.47 × 10^−3^	114.24%
CL_2_	L/h	2.34 × 10^−3^	2309.27%	2.36 × 10^−4^	3499.48%
V_2_	L	9.07 × 10^−5^	24,905.54%	2.91 × 10^−3^	5199.34%
CL_3_	L/h	5.43 × 10^−6^	10,224.64%	8.30 × 10^−5^	10,381.48%
V_3_	L	6.03 × 10^−3^	557.66%	2.22 × 10^−3^	5648.0%
CL/kg	L/h/kg	1.58 × 10^−4^	1407.07%	2.24 × 10^−4^	445.91%
Vc/kg	L/kg	6.96 × 10^−3^	1467.27%	6.49 × 10^−3^	114.24%
CL2/kg	L/h/kg	0.011	2309.27%	1.04 × 10^−3^	3499.48%
V2/kg	L/kg	4.09 × 10^−4^	24,905.54%	0.013	5199.34%
CL3/kg	L/h/kg	2.44 × 10^−5^	10,224.64%	3.67 × 10^−4^	10,381.48%
V3/kg	L/kg	0.027	557.66%	9.80 × 10^−3^	5648.00%
A	µg/mL	60.72	-	888.9	-
B	µg/mL	1026.6	-	36.07	-
C	µg/mL	5.348	-	199.7	-
K_α_	1/h	27.31	-	0.315	-
K_β_	1/h	0.025	-	0.048	-
K_γ_	1/h	7.75 × 10^−4^	-	6.86 × 10^−3^	-
K_10_	1/h	0.023	2032.92%	0.034	460.31%
K_12_	1/h	1.515	2735.99%	0.161	3501.35%
K_21_	1/h	25.79	25,012.37%	0.081	6267.34%
K_13_	1/h	3.51 × 10^−3^	10,329.38%	0.057	10,382.11%
K_31_	1/h	8.99 × 10^−4^	10,239.84%	0.037	11,818.42%
C_max_	µg/mL/mg Dose	704.4	-	681.6	
C_0_	µg/mL	1092.6	-	1124.7	-
t_1/2_	h	894.8	0.0%	101	0.00%
R^2^	-	0.418	-	0.5752	-

Model parameters: Akaike information criterion (AIC) = (#Pts) × Log(Obj) + 2(#Parameters) = −3.3697; Schwarz criterion (SC) = (#Pts) × Log(Obj) + (#Parameters) × (Log(#Pts)) = 9.1963; optimization time: 2.292969 s; total simulations: 3288; weighted sum of squared errors = 7.7402^−1^; weighting: 1/Yhat^2^.

## Data Availability

The data presented in this study are available upon request from the corresponding author due to related studies that are ongoing and pending patent applications.

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
