# Peer review of "Pharmacokinetics of Levofloxacin Entrapped in Non-Ionic Surfactant Vesicles (Niosomes) in Sprague Dawley Rats"

_pharmaceutics, 2025, doi:10.3390/pharmaceutics17020275_

Round 1
Reviewer 1 Report
Comments and Suggestions for Authors
1. Data should be kept with significant digits in Tables, like Table 5. Do not keep all digits.
2. Figures 7 & 8, please use mean +/- SD (with error bar) for data presentaion.
3. Do not use "^-05" for data presentaion. Please use "x 10-5" (-5 is superscript).
4. Figure 10, please add the error bar.
5. Table 10 should be deleted. This is raw data. Just mention the results in th text.
6. Abstract, Introduction and Conclusion should be shorten.
Author Response
- Data should be kept with significant digits in Tables, like Table 5. Do not keep all digits. Response: Done for all Tables
2. Figures 7 & 8, please use mean +/- SD (with error bar) for data presentation.
Response: Figures 7 & 8 were generated by modeling software (Gastroplus, SimulationsPlus, CA) and has been presented in conventional format for presenting such data.
3. Do not use "^-05" for data presentation. Please use "x 10-5" (-5 is superscript).
Response: Done. All exponential data have been represented by x10n
4. Figure 10, please add the error bar.
Response: Done. Error bar (+/- SD) added to Figure 10
5. Table 10 should be deleted. This is raw data. Just mention the results in the text.
Response: Done. Table 10 has been deleted and referenced in the manuscript as recommended.
6. Abstract, Introduction and Conclusion should be shortened
Response: Done. Abstract has been shortened from about 500 words to 426 words, introduction and conclusion have also been compressed while preserving the readability of the manuscript.
Reviewer 2 Report
Comments and Suggestions for Authors
pharmaceutics-3420999
Pharmacokinetics of Levofloxacin entrapped in non-ionic surfactant vesicles (niosomes) in Sprague Dawley rats
Amusa S Adebayo *, Satish Jankie, Jenelle Johnson, Lexley Pinto Pereira, Kafilat Agbaje, Simeon K. Adesina
The aim was to determine the pharmacokinetic parameters and any impact of niosomes on in vivo distribution of levofloxacin in rats. This theme is in the scope of Pharmaceutics. The data obtained are of interest for researchers.
However, I have a number of questions and comments.
The abstract should indicate the composition of the niosomes. It is necessary to provide the dimensionality of the zeta potential and the hydrodynamic radius (or diameter?) of the particles in the same part.
In the Materials and Methods par., indicate the origin and purity of the reagents used to form niosomes.
Why are the abbreviations LEV and CLA given in the note to Table 1, which are not used in it? The zeta potential dimension should be given in the table header, and it should be clarified whether the particle diameter or radius values ​​are given. The EE values ​​are given with excessive precision. Check the number of decimal places in other tables as well. In what units is the DCP concentration given?
In my opinion, the data from scanning electron microscope, transmission electron microscopy and DLS of levofloxacin niosomes are in poor agreement with each other.
It would be useful to provide characteristics of particles unloaded with levofloxacin. What is the reason for such high negative zeta potential values? Why was composition N11 chosen as optimal? I believe that the introduction should pay attention to the question of the possibilities that arise when modifying niosomes with additives of ionic surfactants.
The curve in Fig. 6 reproduces the line from Fig. 6 of reference 31. It would be useful to investigate and discuss the release rate of LEV from niosomes of different compositions.
I recommend deleting Fig. 10 and give the mean residence time values ​​ in the text
Minor notes.
The manuscript should be edited more carefully. The presentation of the obtained data should be brought into uniformity. For example, in Table 9, the fractional part of the numbers is separated by both a period and a comma.
Delete the repeated phrases: lines 274 and 405.
The numbering of the figures in the text on lines 458-466 should be clarified.
I recommend making more complete captions to the figures and tables: adding information about the composition of niosomes for which data are provided (Fig. 6, 9), explaining the multi-colored symbols in Fig. 7 and 8, etc.
The figures should be designed in a uniform manner.
Taking into account the above, I believe that manuscript needs major revision before it can be recommended for publication
Author Response
The aim was to determine the pharmacokinetic parameters and any impact of niosomes on in vivo distribution of levofloxacin in rats. This theme is in the scope of Pharmaceutics. The data obtained are of interest for researchers.
However, I have a number of questions and comments.
Comment 1: The abstract should indicate the composition of the niosomes. It is necessary to provide the dimensionality of the zeta potential and the hydrodynamic radius (or diameter?) of the particles in the same part.
Response:
Niosomes composition, hydrodynamic diameter and Zeta potential have been incorporated into Abstract.
In the Materials and Methods par., indicate the origin and purity of the reagents used to form niosomes.
Response: The sources of materials and reagents have been incorporated in the manuscript, lines 90 - 92
Why are the abbreviations LEV and CLA given in the note to Table 1, which are not used in it? The zeta potential dimension should be given in the table header, and it should be clarified whether the particle diameter or radius values ​​are given. The EE values ​​are given with excessive precision. Check the number of decimal places in other tables as well. In what units is the DCP concentration given?
Response: LEV and CLA have been corrected accordingly. Dimension of Zeta potentials (mV) has been added) and the vesicle size has been defined as the hydrodynamic diameter (nm).
In my opinion, the data from scanning electron microscope, transmission electron microscopy and DLS of levofloxacin niosomes are in poor agreement with each other.
Response: This may be because SEM data was from lyophilized powder while DNS from nanosizer was from colloidal dispersion of niosomes, as required by the instruments. Some aggregation of vesicles after freeze drying is responsible for the larger size brought to the fore.
It would be useful to provide characteristics of particles unloaded with levofloxacin. What is the reason for such high negative zeta potential values? Why was composition N11 chosen as optimal? I believe that the introduction should pay attention to the question of the possibilities that arise when modifying niosomes with additives of ionic surfactants.
Response: The Zeta potential is dependent on proportions of Cholesterol and dihexacetyl phosphate; Span 60 is non-ionic and carries no charge. Levofloxacin is an acidic drug with available -COOH group. The preponderance of negatively charged entities in the formulation may account for the overal pattern of Zeta potential seen. Due to electrostatic repulsion, high Zeta potential values (negative or positive) is desirable for electrostatic repulsion that would prevent niosomes fusion and particle growth on collision.
Formulation 11 of screening experiment fell within +/- 2 sigma of optimizer range but the actual formulation used for animal study was prepared using DoE recommended factors combination and has been included in the manuscript accordingly.
The curve in Fig. 6 reproduces the line from Fig. 6 of reference 31. It would be useful to investigate and discuss the release rate of LEV from niosomes of different compositions.
Response: Dissolution data on a formulation from screening and an optimized formulations have been plotted in Figure 6, lines 354 - 355.
I recommend deleting Fig. 10 and give the mean residence time values ​​ in the text
Response: Another peer-reviewer has recommended including erro bars on Figure 10. We have left the Fig. 10 for editor's final decision.
Minor notes.
The manuscript should be edited more carefully. The presentation of the obtained data should be brought into uniformity. For example, in Table 9, the fractional part of the numbers is separated by both a period and a comma.
Response: Done. Data has been re-presented as requested in unified formats
Delete the repeated phrases: lines 274 and 405.
Response:
The phrase has been deleted.
The numbering of the figures in the text on lines 458-466 should be clarified.
Response: Figure numbers have been corrected.
I recommend making more complete captions to the figures and tables: adding information about the composition of niosomes for which data are provided (Fig. 6, 9), explaining the multi-colored symbols in Fig. 7 and 8, etc.
Response:
Informations has been added to figures, multi-colored line has been explained in the manuscript.
The figures should be designed in a uniform manner.
Response: All figures prepared in MS Excel have similar format. Others were generated with the Gastroplus (PK Modeling & Simulations software) as part of the model output. Important gfeatures will be lost if presented in MS Excel format.
Taking into account the above, I believe that manuscript needs major revision before it can be recommended for publication

Reviewer 3 Report
Comments and Suggestions for Authors
The article's positive differential lies in the combination of a solid technical approach (DoE, HPLC, PK modeling) with a relevant clinical focus (improvement in the biodistribution and efficacy of levofloxacin). The work contributes significantly to developing controlled release systems, especially in antibiotics.
But I suggest some content improvement.
The introduction mentions multiple contexts without fluid organization; reorganization is suggested to focus first on the background on niosomes.
Methodology:
Some data in tables lack sufficient justification or analysis. Example: Table 6 and its AUC and MRT data are described, but there is no clear discussion of their statistical relevance (lines 368–386)​.
The sample calculation (n=6 per group) is not justified in the text.
Results:
Detailed graphs and tables are presented (lines 296–303, Figures 3A–C). However, the correlation between results and hypotheses is not explicitly discussed in some sections.
Suggestion: Add clear explanations for the high standard deviations in certain data, such as the coefficient of variation in Table 7 (lines 368–371)​.
Discussion:
The pharmacokinetic impact of encapsulation is addressed, but there is no direct link to clinical benefits in quantitative terms (lines 394–435)​.
The authors should emphasize the clinical potential of encapsulation in specific levofloxacin-resistant bacteria.
Conclusion: this is consistent, but does not sufficiently explore the limitations of the study.
Author Response
The introduction mentions multiple contexts without fluid organization; reorganization is suggested to focus first on the background on niosomes.
Response: The introduction has been reorganized. Starting with niosomes as delivery system, we progressed to levofloxacin and why niosomes formulation will be helpful in resistance mitigation to in vivo PK study that is the focus of the current study.
Methodology:
Some data in tables lack sufficient justification or analysis. Example: Table 6 and its AUC and MRT data are described, but there is no clear discussion of their statistical relevance (lines 368–386)​.
Response:
The PK parameters in previous Table 6 (Now Table 7, due to addition of optimizer result in response to another reviewer) have been explained under results, lines 358 - 366.
The sample calculation (n=6 per group) is not justified in the text.
Response:
Statistically, a minimum of 3 samples are needed to enable stadnard deviation calculations. The larger the sample size, the more representative the data. However, the principle of 3R's (Reduction, Refinement and Replacement) i.e. Reduction: Minimize the number of animals used; Refinement: Employ techniques that reduce pain and distress; Replacement: Substitute animal with nonanimal or animals that are less sentient or lower on the phylogenetic scale guided the Animal Ethics Committee in approving the study protocol. Applying these principle enable apprroval of 12 rats and 6 rats were used for niosomes and 6 for levofloxacing injection (control). Thiss has been incorporated under "Design of Bioavailability studies" lines 165 - 166.
Results:
Detailed graphs and tables are presented (lines 296–303, Figures 3A–C). However, the correlation between results and hypotheses is not explicitly discussed in some sections.
Response:
Suggestion: Add clear explanations for the high standard deviations in certain data, such as the coefficient of variation in Table 7 (lines 368–371)​.
Response:
CV values are used to evaluate model reliability. The higher the CV, the less representative of the data is the model producing those CV values. Hogh CV is a feature of biological experiments due to int=herent high variability of the organic systems. This is why PK Moddels are used to evaluate data using established algorithms that could partition overaall data into treatment effect and error (random) effects. As can be seen, CV increases from Non-compartmental to 3-compartmental models for applicable parameters. R2 accompanying parameters are important in showing how predictive the dependent variables (i.e. Cp) is to independent variable (i.e., time)
Discussion:
The pharmacokinetic impact of encapsulation is addressed, but there is no direct link to clinical benefits in quantitative terms (lines 394–435)​.
Response:
The clinical potential has been incorporated under discussion section, lines 472 to 486.
The authors should emphasize the clinical potential of encapsulation in specific levofloxacin-resistant bacteria.
Response:
This has been incorporated under conclusions, line 497 - 500) as summarized here "... as was previously reported [31] (Jankie, et al, 2012), a 50 % reduction in MIC/MBC of some fluoroquinolones niosomes (including levofloxacin, gatifloxacin and ciprofloxacin) on drug-resistant strains of P. aeruginosa, E. coli and S. aureus could enable dose reduction... "
Conclusion: this is consistent but does not sufficiently explore the limitations of the study.
Response:
The limitations have been paraphrased in last line of conclusion section as "Furthers studies are needed to confirm intracellular delivery and the delivery mechanisms of niosomes-encapsulated levofloxacin into multi-drug-resistant bacterial cells". Cargo-carying niosomes plus targeting ligands are currently being investigated in our lab and will be published in due course.
Round 2
Reviewer 2 Report
Comments and Suggestions for Authors
In accordance with the reviewer's comments, authors made a number of technical corrections to the manuscript. However, I believe that the questions regarding the differences in data from scanning electron microscope, transmission electron microscopy and DLS, deserve a more detailed answer and should be reflected in the article. The same applies to the question of the possibilities that arise when modifying niosomes with additives of ionic surfactants.
In addition, the authors should have heeded the reviewer's advice to unify the figures presented. For example, in Figures 7 the indices “e” and “E” are used, the number of digits on the axes varies greatly, etc. In addition, the authors should have listened to the reviewer's advice to unify the presented figures. Once again, I ask the authors to check the need to provide the obtained values ​​with such precision. For example, Table 1 provides the value EE 12.39788
Taking into account the above, I believe that manuscript needs minor revision before it can be recommended for publication
Author Response
Pharmaceutics – Rat PK Review Second Round
Comments and Suggestions for Authors
In accordance with the reviewer's comments, authors made a number of technical corrections to the manuscript. However, I believe that the questions regarding the differences in data from scanning electron microscope, transmission electron microscopy and DLS, deserve a more detailed answer and should be reflected in the article. The same applies to the question of the possibilities that arise when modifying niosomes with additives of ionic surfactants.
Response: Our sincere apology for the oversight of the Reviewer’s comment in our previous response. Typical DLS figures and data corresponding to sample for scanning and transmission electron microscopy have been incorporated in the manuscript. The Zeta potential profile and statistical data corresponding to the dicetyl phosphate (DCP, the ionic surfactant) level used as charge inducer for optimized formulation have been incorporated in the manuscript. However, lyophilizedd powder of niosomes were used for SEM imagiing coontrast to the colloiidal ssuspension used for TEM imaging. Agglomeratiion and clumping on lyophilization is responsible for the observed ddifferences in SEM and TEM images, especiallly the partcle size.
In addition, the authors should have heeded the reviewer's advice to unify the figures presented. For example, in Figures 7 the indices “e” and “E” are used, the number of digits on the axes varies greatly, etc. In addition, the authors should have listened to the reviewer's advice to unify the presented figures. Once again, I ask the authors to check the need to provide the obtained values ​​with such precision. For example, Table 1 provides the value EE 12.39788
Responses:
Figures presentation have been unified as recommended. The number of digit as presented in Figures 7 and 8 (A), (B) and (C) respectively depends of model quantitative values that were generated by the Gastroplus modeling software. Altering the visualization will require turning the graph to MS Excel format will defeat the visualization power of the models.
We thank the Reviewer for their time and attention.

Reviewer 3 Report
Comments and Suggestions for Authors
no comments
Author Response
Dear Edditor,
The reviewer seems to be satisfied with our responses and has o.k. the various questions below. We thank the reviewer for their time and attention.
Sincerely,
Dr. Adebayo